# TRPC Channels in Proteinuric Kidney Diseases

**DOI:** 10.3390/cells9010044

**Published:** 2019-12-23

**Authors:** Gentzon Hall, Liming Wang, Robert F. Spurney

**Affiliations:** Division of Nephrology, Department of Medicine, Duke University; Durham, NC 27710, USA; gentzon.hall@duke.edu (G.H.); liming.wang@duke.edu (L.W.)

**Keywords:** transient receptor potential cation channel, transient receptor potential canonical channel, TRPC channel, focal segmental glomerulosclerosis, diabetic kidney disease, diabetic nephropathy, chronic kidney disease, TRPC6, TRPC5, TRPC3

## Abstract

Over a decade ago, mutations in the gene encoding *TRPC6* (transient receptor potential cation channel, subfamily C, member 6) were linked to development of familial forms of nephrosis. Since this discovery, TRPC6 has been implicated in the pathophysiology of non-genetic forms of kidney disease including focal segmental glomerulosclerosis (FSGS), diabetic nephropathy, immune-mediated kidney diseases, and renal fibrosis. On the basis of these findings, TRPC6 has become an important target for the development of therapeutic agents to treat diverse kidney diseases. Although TRPC6 has been a major focus for drug discovery, more recent studies suggest that other TRPC family members play a role in the pathogenesis of glomerular disease processes and chronic kidney disease (CKD). This review highlights the data implicating TRPC6 and other TRPC family members in both genetic and non-genetic forms of kidney disease, focusing on TRPC3, TRPC5, and TRPC6 in a cell type (glomerular podocytes) that plays a key role in proteinuric kidney diseases.

## 1. The Burden of Chronic Kidney Disease (CKD)

CKD is a public health problem that affects more than 20 million Americans in the United States [1]. The disease is caused by a diverse group of primary, secondary, and genetic disorders that impair the ability of the kidney to remove waste, regulate intravascular volume, reclaim essential nutrients, and stimulate erythropoiesis [2]. As the disease progresses, these essential functions become increasingly impaired, eventually leading to end-stage kidney disease (ESKD). Of the ≈20 million Americans with CKD, almost 600,000 have irreversible kidney failure and require either renal transplantation or dialysis to sustain life [3]. The expense of providing ESKD care to this patient population costs ≈50 billion dollars per year in the United States [3]. Moreover, morbidity and mortality are markedly increased in CKD patients with lesser degrees of renal impairment, which is largely caused by an increased risk of cardiovascular events as renal function declines [1,4]. Although current treatments for CKD may slow disease progression, many patients progress to more advanced stages of CKD, and ESKD, despite available therapies [2]. As a result, there is a significant interest in developing new treatment approaches.

## 2. The Glomerular Podocyte Plays a Key Role in Proteinuric Kidney Diseases

Podocytes are highly differentiated, postmitotic cells that play a critical role in maintaining the integrity of the glomerular filtration barrier [5]. As shown in Figure 1A,B, their interdigitating foot processes (FPs) cover the exterior surface of the glomerular basement membrane (GBM), and form a specialized intercellular junction between adjacent FPs known as the slit diaphragm (SD) [6]. The SD is a sieve-like, size-selective filtration barrier, which prevents loss of cells and macromolecules into the glomerular ultrafiltrate [7,8]. It is composed of proteins commonly found in tight and adherens junctions such as P-cadherin, zonula occludens-1 (ZO-1), and catenin family members, as well as proteins predominantly expressed in podocytes including nephrin, neph1, and podocin [7,8]. Both nephrin and neph1 are members of the immunoglobulin (Ig) superfamily [7]. Their extracellular domains contain multiple IgG-like motifs, which can aggregate and form zipper-like structures that surround the glomerular capillaries, creating a sieve-like filtration barrier. Both nephrin and neph1 also have a single transmembrane domain and a short intracellular carboxy-terminus, which serves as a scaffold for assembling signaling complexes with other podocyte proteins such as CD2AP (CD2-associated protein) and podocin [8,9]. The carboxy-terminus of nephrin has conserved tyrosine residues that can be phosphorylated by Src family kinases [7,10,11,12,13]. Phosphorylated nephrin recruits additional signaling molecules to the scaffolding complex that, in turn, influence multiple downstream signaling pathways and modulate cytoskeletal reorganization, cellular survival, membrane trafficking, cellular adhesion, and mechano-signaling [7,10,14,15].

In addition to their role in forming a size selective filtration barrier, podocytes stabilize the glomerular architecture by counteracting the elastic forces distending the GBM [6]. Indeed, the hydraulic pressure within glomerular capillaries is unusually high compared to other capillary beds [16]. It is estimated that podocytes are responsible for approximately 40% of the hydraulic resistance of the filtration barrier [6]. Moreover, accumulating evidence suggests that dynamic signaling between the SD and focal adhesions may promote both podocyte adherence to the GBM and maintenance of glomerular filtration barrier function [7,17,18]. In support of this hypothesis, lamellipodia formation at focal adhesions are necessary for cell migration and are induced downstream of nephrin phosphorylation in cultured podocytes [19,20]; conversely, β1 integrin activation induces tyrosine phosphorylation of nephrin [21]. This crosstalk between the SD and focal adhesions may help podocytes adapt to changes in the microenvironment including alterations in glomerular filtration rate or glomerular pressure. In addition, disruption of these signaling pathways in glomerular diseases may contribute to the loss of glomerular filtration barrier function, causing proteinuria. As the disease progresses, podocytes detach from the GBM due either to reduced cellular adhesion or cell death. Because podocytes are terminally differentiated, postmitotic cells, podocytes that are lost, cannot be effectively replaced [22,23]. As a result, a decrease in podocyte number of greater than ≈20–40% [24,25] leads to the collapse of the glomerular tuft, glomerulosclerosis, and disease progression [22,23]. On the basis of these observations, the current review focuses on the role of podocyte TRPC channels in proteinuric kidney diseases, but we acknowledge that expression of TRPC channels in other kidney cell types are likely to play a role in these disease processes.

## 3. TRPC Family Members

TRPC channels are part of a larger TRP family, which was first discovered in *Drosophila melanogaster* [26,27]. As shown in Figure 1C, the vertebrate TRPC family has seven members, which can be divided into four subgroups: TRPC1, TRPC2, TRPC4/5, and TRPC3/6/7 (G) [26,27]. In humans, TRPC2 is a pseudogene, although the channel plays a role in pheromone signaling in rodents [26]. All TRPC ion channels are calcium permeable, but the channels are poorly selective, with permeability ratios (PCa/PNa) that vary significantly between family members [26]. The calcium influx is stimulated following receptor-induced phospholipase C (PLC) activation in response to both tyrosine kinase receptors (TKRs) and G protein coupled receptors (GPCRs) [27]. PLC catalyzes hydrolysis of phosphatidylinositol 4,5-bisphosphate (PIP2) and generates the second messengers diacylglycerol (DAG) and inositol 1,4,5 trisphosphate (IP3) [28] that differentially affect TRPC activation, as reviewed elsewhere [27].

Several members of the TRPC family may be activated by additional mechanisms that are distinct from receptor operated channel (ROC) activation. For example, mechanical stretch was reported to activate TRPC1 in 2005 [29]. More recent studies suggest that other TRPC family members may also be responsive to mechanical stress, including TRPC3, TRPC5, and TRPC6 [30,31,32,33,34,35]. For example, TRPC6 was found to induce cationic currents in podocytes following mechanical stimulation, which persisted in the presence of either PLC inhibitors or blockade of G-protein activation, but was inhibited by TRPC6 knockdown as well as by pharmacologic blockade of TRPC6 [36]. Moreover, a point mutation in TRPC6 eliminates mechanical activation of cationic currents in podocytes without affecting activation by GPCRs or cell-permeable DAG analogs [34]. Although the role of mechanical stretch in activation of TRPC family members has recently been questioned [37], these data are consistent with the notion that stretch-induced TRPC6 activation in podocytes is mediated by both receptor-dependent and receptor-independent mechanisms.

A third mechanism of TRPC activation is oxidative stress, which is reported to activate TRPC5 and TRPC6 [38,39,40]. In podocytes, TRPC6 activation by angiotensin II and cell permeable DAG analogs is inhibited by both scavengers of reactive oxygen species (ROS) and by pharmacologic inhibition of NADPH oxidase 2 (NOX2) [41,42]. Similarly, ROS quenchers inhibit TRPC6 activation by ATP [43]. In addition, TRPC6 coimmunoprecipitates with the catalytic subunit of NOX2, and the TRPC6–NOX2 interaction appears to require podocin because knockdown of the scaffolding protein podocin eliminates the TRPC6–NOX2 protein–protein interaction [42]. Taken together, these data suggest that localized production of ROS plays a key role in GPCR-induced TRPC6 activation.

## 4. TRPC6 in Familial Forms of Nephrosis

In 2005, Winn et al. identified a point mutation in *TRPC6*, which segregated with disease in a large family with autosomal dominant focal segmental glomerulosclerosis (FSGS) [44]. In heterologous expression systems, the mutant channel exhibited a gain-of-function in response to angiotensin II and to cell permeable DAG analogs [35,44]. Shortly after this initial report, Pollak and coworkers described five additional families with autosomal dominant FSGS caused by mutations in the *TRPC6* gene [45]. In this study, immuno-electron microscopy localized podocyte TRPC6 to the cell body, major processes, and foot processes adjacent to the slit diaphragm [45]. This group further demonstrated that TRPC6 co-immunoprecipitated with nephrin and podocin [45,46], raising the possibility that TRPC6, nephrin, and podocin physically interact and might form a signaling complex with TRPC6 at the SD [14,35,46]. In support of this view, the interaction between the cytosolic domains of TRPC6 and podocin regulate TRPC6 activity [35,36,45,46]. Moreover, podocyte FPs contain a contractile apparatus [45,47,48] that may be regulated by changes in calcium levels within the microenvironment [45,48]. These observations fit nicely with a role for TRPC6 as a mechanosensor that directly interacts with, and perhaps influences, the activity of the nephrin/podocin/CD2AP signaling complex at the SD.

Since the initial reports, multiple mutations in the *TRPC6* gene have been linked to familial forms of FSGS [49,50,51,52,53,54,55,56,57]. Examining the clinical characteristics of patients with these mutations has provided insights into the role of TRPC6 in glomerular diseases. For example, the majority of TRPC6 mutations causing FSGS promote a gain of channel function [49,50,51,52,53,54,55,56,57]. These data suggest that excessive intracellular calcium levels contribute to the pathophysiology of the disease in this patient population. Consistent with this hypothesis, a subset of these activating mutations causes large increases in channel currents, which leads to a more aggressive phenotype, with patients presenting in early childhood [54]. The majority of FSGS patients, however, with TRPC6 mutations present as young adults [49], although there is a wide age range between family members that carry the same causative mutation [49]. For example, in the large pedigree reported by Winn et al. [58], family members carrying the TRPC6 mutation presented at a mean age of 33, with an age range at presentation from 16 to 61 years of age. In a more recent study by Zhu et al. [55], the proband presented at age 35, and his sister was diagnosed with FSGS at age 41, but their father with the same activating mutation had normal renal function at age 75. Although the magnitude of the gain of function may lead to a more aggressive phenotype [54], it has been suggested that, in addition to the presence of the causative TRPC6 mutation, the development of disease may be precipitated by a “second hit” [59,60]. In support of this hypothesis, we developed a transgenic (TG) mouse model that promoted continuous activation of endogenous TRPC6, specifically in podocytes [61]. Surprisingly, this model did not exhibit a kidney phenotype at baseline. However, treatment with the podocyte toxin puromycin aminonucleoside (PAN) induced heavy proteinuria and glomerulosclerosis in TG mice, whereas PAN induced only mild disease in wild type mice [61]. Thus, a “second hit” with the podocyte toxin PAN induced a severe phenotype in the presence of persistent TRPC6 activation in the TG animals.

Although the majority of TRPC6 mutations that cause FSGS are activating, ≈25% of the reported mutations cause a loss of function [49], including a novel TRPC6 mutation (G757D) that acts as a dominant negative [49]. In contrast to patients with activating mutations (see above), the age at presentation for inactivating mutations tends to be in early childhood [49,50,51,52,53]. For example, the two patients with the dominant negative mutation (G757D) presented in early infancy [49]. In addition, most of these mutations are heterozygous missense mutations, which have been described in children with non-familial forms of steroid-resistant nephrotic syndrome [49,50,51,52,53]. These data suggest that some basal level of TRPC6 activity may be required to maintain normal glomerular function, which may be particularly important in early childhood.

## 5. TRPC Family Members in Glomerular Diseases: Mechanisms of Renal Injury

Familial forms of kidney disease account for only a small percentage of patients with CKD. The vast majority of kidney disease is caused by systemic disease processes, including diabetes mellitus; hypertension; autoimmune diseases such as lupus erythematosus; and, to a lesser extent, primary glomerular diseases including non-familial forms of FSGS, membranous glomerulonephritis (MGN), and minimal change disease (MCD). A role for TRPC6 in primary glomerular disease processes was first suggested by Reiser and colleagues [62], who demonstrated that TRPC6 was upregulated in microdissected glomeruli in human kidney biopsy specimens from patients with MGN and MCD, with a similar trend in patients with FSGS. Upregulation of TRPC6 in glomerular diseases likely plays a role in disease pathogenesis because overexpression of TRPC6 in heterologous expression systems causes enhanced intracellular calcium levels following TRPC6 activation compared to cells expressing their endogenous complement of TRPC6 channels [53,54]. These data suggest that enhanced expression of wild type TRPC6 channels in acquired proteinuric kidney diseases promote a similar pathology (enhanced intracellular calcium levels) as TRPC6 gain-of-function mutations in familial forms of FSGS. In support of this possibility, podocyte-specific overexpression of TRPC6 in vivo promotes proteinuria and modest glomerulosclerosis [63]. Since these initial observations, enhanced expression of TRPC6 has been demonstrated in other kidney diseases including rodent models of diabetic nephropathy [61,64,65,66,67,68,69], nephrotoxic serum nephritis [70,71], rodent models of FSGS [61,64,72,73,74], and ureteral obstruction [75,76]. Thus, increased expression of TRPC6 may be a common finding in a broad range of pathologic conditions affecting the kidney.

The molecular mechanisms regulating TRPC6 expression have been investigated in both renal and extrarenal disease processes [61,64,72,74,77,78]. An important observation was that activation of TRPC6 stimulated a positive feedback loop that promoted further expression of TRPC6 in the heart [77,78,79]. As shown in Figure 2, PLC-coupled receptors, such as GPCRs for angiotensin II or endothelin 1, activate TRPC6, which promotes calcium entry into the cell and stimulates the calcium-sensitive phosphatase calcineurin [80]. An important calcineurin substrate is the family of NFAT (nuclear factor of activated T cells) transcription factors [80]. NFAT family members were originally discovered in cells of the lymphoid lineage, but NFAT isoforms are also expressed in non-immune cells, with some family members expressed ubiquitously [81]. In quiescent cells, NFAT isoforms are phosphorylated and located in the cytoplasm [81,82]. Calcineurin dephosphorylates NFAT, which permits translocation to the nucleus and stimulation of gene transcription. An important gene target of calcineurin is the ion channel TRPC6 [61,74,77]. As shown in Figure 2, this pathway creates a positive feedback loop characterized by enhanced intracellular calcium levels, stimulation of calcineurin-NFAT signaling, and induction of TRPC6, which further augments intracellular calcium levels. This positive feedback pathway plays a key role in cardiovascular disease processes such as pathologic cardiac hypertrophy [77,78,83] and, more recently, this signaling pathway has been demonstrated to play a role in kidney diseases [61,84,85], including animal models of FSGS [61,74,84] and diabetic kidney disease [65,66]. Indeed, studies by Wang et al. [84] found that podocyte-specific expression of a constitutive active NFAT isoform caused glomerulosclerosis resembling FSGS.

In addition to NFAT, calcineurin has numerous substrates including transcription factors, receptors, ion channels, cytoskeletal proteins, targeting proteins, and proteins involved in apoptotic pathways [86]. Thus, TRPC6-induced calcineurin activation affects numerous downstream signaling pathways that may contribute to disease progression in CKD. For example, the podocyte protein synaptopodin associates with the actin cytoskeleton and plays a key role in maintaining the complex morphology of the glomerular podocyte, as well as inhibiting cell surface expression of TRPC6 [87]. As shown in Figure 3, Faul et al. [88] found that synaptopodin was phosphorylated by either protein kinase A (PKA) or calcium-dependent protein kinase II (CaMKII). Phosphorylation of synaptopodin provides a docking site for 14-3-3 proteins and prevents degradation of synaptopodin by the cysteine peptidase cathepsin L [88]. Dephosphorylation of the 14-3-3 docking site by calcineurin promotes synaptopodin degradation, destabilizes the actin cytoskeleton, and disrupts the integrity of the glomerular filtration barrier.

As discussed above, podocytes have a limited capacity for proliferation [5,6]. As a result, a decrease in podocyte number leads to collapse of the glomerular tuft, glomerulosclerosis, and disease progression [5,6,22,23]. TRPC6-induced calcium/calcineurin/NFAT signaling enhances podocyte loss by inducing podocyte apoptosis [84,89,90]. As shown in Figure 3, mechanisms of calcineurin-induced apoptosis include stimulating mitochondrial fragmentation by Drp1 (dynamin-related protein 1) [91] and activation of the apoptosis inducing Bcl-2 family member BAD (Bcl-2-associated death promoter) [92]. As shown in Figure 3, Drp1 is phosphorylated and inhibited by PKA [91], and BAD is phosphorylated by Akt, which causes sequestration of BAD by 14-3-3 proteins [92]. Dephosphorylation of both proteins by calcineurin induces apoptosis [91,92]), and both Drp1- and BAD-dependent podocyte apoptosis have been implicated in CKD progression [15,93,94,95].

TRPC6 activation also contributes to the pathogenesis of glomerular diseases by mechanisms that are independent of calcineurin activation. For example, calcium influx induced by TRPC6 activates the cysteine protease calpain [96], as shown in Figure 3. Recent studies suggest that TRPC6 binds calpain-1 and calpain-2, which causes membrane localization of calpain isoforms and promotes their activation [97]. Calpains have numerous downstream substrates including cytoskeletal proteins [97,98] such as the large cytoskeletal protein talin-1, which links integrins to the actin cytoskeleton [99]. Knockout of talin-1 specifically in glomerular podocytes causes severe proteinuria [99], and TRPC6-induced calcium influx increases calpain-1 activity [96]. These observations are relevant to proteinuric kidney diseases because calpain-induced cleavage of talin-1 is increased in animal models of glomerular disease, and urinary calpain activity is increased in patients with FSGS and MCD [96,99]. Moreover, calpain inhibitors attenuate proteinuria in animal models [96,99]. Taken together, these data suggest an important role for the calcium sensitive protease calpain in TRPC6-induced kidney injury.

In addition to enhancing intracellular calcium levels, TRPC family members are reported to activate the small GTPases RhoA and Rac1 [100,101,102], and both these Rho GTPase family members are implicated in the pathogenesis of idiopathic and hereditary glomerular diseases [103,104,105,106,107]. Indeed, RhoA and Rac1 play key roles in regulating cytoskeletal dynamics in podocytes, and dysregulated activity of these small GTPases in glomerular disease processes causes proteinuria [103,107]. In this regard, studies by Greka and co-workers found that Rac1 and RhoA associate with, and are activated by, TRPC5 and TRPC6, respectively [100]. Consistent with an important role for TRPC5 in glomerular diseases, additional studies by this same group suggested that inhibition of TRPC5 ameliorated glomerular injury in proteinuric rodent models [108,109], as discussed further below.

Glomerular podocytes are reported to express TRPC1, TRPC3, TRPC4, TRPC5, and TRPC6; however, only TPRC3, TRPC5, and TRPC6 have been shown to contribute to calcium entry in podocytes using electrophysiologic and pharmacologic methodologies [110]. Although TRPC5 and TRPC6 have received the most attention, accumulating evidence suggests that TRPC3 also plays a role in kidney diseases [75]. Similar to other TRPC family members, activation of TRPC3 enhances intracellular calcium levels and stimulates downstream calcium signaling cascades. An important observation was that TRPC3 is upregulated in glomerular diseases in a fashion similar to TRPC6 [61,71,73,111,112], with no change in expression of TRPC5 [61,71,73,112]. Like TRPC6, upregulation of TRPC3 occurs by calcineurin-dependent mechanisms in pathologic processes [78]. This observation may be important in glomerular diseases because increased expression of TRPC3 and TRPC6 in kidney diseases may enhance the relative contributions of these TRPC family members to elevated intracellular calcium levels. Moreover, compensatory upregulation of TRPC3 is observed in vascular smooth muscle cells of TRPC6 knockout mice, which promoted higher basal cation entry and likely contributed to enhanced vascular contractility and increased systemic blood pressure in this model [113]. Similarly, TRPC3 was upregulated in renal cortices of TRPC6 knockout rats under basal conditions [71,73]. Thus, selective blockade of TRPC6 may promote compensatory upregulation of other TRPC family members and may not be sufficient in inhibiting the calcium influx induced by TRPC activation in pathologic processes.

## 6. Targeting TRPC6 to Treat FSGS

Since the discovery that mutations in TRPC6 caused familial forms of FSGS, over 50 genes have been linked to inherited causes of glomerular disease [114]. However, relatively few non-syndromic, genetic causes of nephrosis had been reported in the year (2005) that mutations in *TRPC6* were linked to the development of autosomal dominant FSGS [44]. These mutations included genes encoding nephrin [115], podocin [116], CD2AP [117,118], and α-actinin-4 [119]. At the time, all these proteins were considered structural elements of either the slit diaphragm or cytoskeleton, although data were emerging to suggest that nephrin, podocin, and CD2AP formed a novel signaling complex at the SD (see above). In contrast, TRPC6 was a classical signaling molecule, an ion channel, which might play an important role in non-familial forms of glomerular disease. As a result, TRPC6 was immediately recognized as an important target for drug development.

To examine the potential therapeutic benefits of targeting TRPC6 in glomerular diseases, Winn and colleagues [111] examined the effects of infusing angiotensin II in whole-body TRPC6 knockout mice [113]. In these experiments, TRPC6-deficient mice developed less albuminuria, as well as a trend toward improved glomerular histology during the 4 week angiotensin II infusion period. Because mice treated with angiotensin II develop severe hypertension, this model evaluated both the indirect effects systemic high blood pressure on the kidney, as well as the direct effects of angiotensin II on the renal tubular, glomerular, vascular, and interstitial compartments of the kidney. Surprisingly, there was no significant effect of TRPC6 knockout on systemic blood pressure at baseline or after angiotensin II infusion using either tail cuff manometry or the “gold standard” for monitoring systemic blood pressure, radiotelemetry [111]. The lack of a difference in blood pressure at baseline was unanticipated, as previous studies had found a modest increase in systemic blood pressure in the knockout mice, as discussed above [113]; however, differences in the genetic backgrounds used in the two studies may have contributed to the differing results [111,113]. Importantly, because Winn and colleagues found that blood pressure was not affected in mice lacking TRPC6, these studies were some of the first observations to suggest that inhibition of TRPC6 had direct protective effects in the kidney.

To investigate in the role of TRPC6 in a glomerular disease process, our laboratory [61] developed a TG mouse that expressed a constitutively active Gq α-subunit specifically in podocytes (Gq mice). Gq links GPCRs such as the receptor for angiotensin II to PLC and activates TRPC6. Thus, this model caused continuous activation of TRPC6. Surprisingly, this TG mouse had no kidney phenotype at baseline, which is reminiscent of humans expressing gain-of-function mutations in TRPC6. We therefore determined if a “second hit” would induce more severe disease, as has been suggested for familial forms of FSGS due to TRPC6 mutations [59,60]. For the studies, TG animals were treated with the podocyte toxin PAN, which normally produces only mild disease in wild type mice. In Gq mice, however, treatment with PAN caused severe nephrosis, prominent tubulointerstitial disease, and glomerulosclerosis resembling the lesions observed in humans with FSGS [61], as shown in Figure 4. Similarly, Krall et al. found that podocyte-specific overexpression of wild type or activating TRPC6 mutants caused mild proteinuria and modest glomerular changes [63]. Although different genetic backgrounds were used in our studies [61] and the Krall et al. study [63], the baseline phenotypes in both studies were either normal or mild, and it would have been of interest to determine if a “second hit” augmented the phenotype in mice overexpressing the wild type and mutant forms of TRPC6 in the Krall et al. model.

On the basis of our findings in Gq mice, we next crossed our TG animals with the whole body TRPC6 knockout mice and treated the animals with PAN to induce glomerular disease. As shown in Figure 4, we found that knockout of TRPC6 significantly reduced proteinuria, tubule dilation, and casts, and markedly reduced the severity of glomerulosclerosis [61]. More recently, Dryer and coworkers created a whole body TRPC6 knockout model in rats [73]. This model had an advantage in that rats are generally more susceptible to glomerular disease compared to mice, including the development of robust glomerular disease following treatment of wild type rats with PAN. Dryer and coworkers found that both wild type and knockout mice injected with PAN developed similar levels of heavy proteinuria during the early phase of the disease, but the albuminuria was significantly reduced in the knockout mice during the later, more chronic phase of the disease process. This reduction in proteinuria was associated with a significant improvement in glomerulosclerosis, tubular damage, interstitial inflammation, and fibrosis. Thus, in both our studies and the study by Dryer and coworkers, whole body knockout of TRPC6 improved the disease process in multiple kidney compartments (glomerular, tubular, and interstitial), perhaps due to beneficial effects of whole body TRPC6 knockout in multiple cell types in the kidney. These data suggest that systemic knockout of TRPC6 has beneficial effects on kidney disease in animal models of FSGS.

Pharmacologic inhibition of either TRPC6 activity or expression has also been shown to ameliorate glomerular disease in animal models of FSGS [64,120]. Although there are currently no selective TRPC6 antagonists in clinical trials, three drug classes commonly used for the treatment of FSGS [121] regulate TRPC6 expression, including angiotensin-converting enzyme inhibitors (ACEIs), angiotensin II receptor blockers (ARBs), and calcineurin inhibitors (CNIs) [61,74,77]. ACEIs and ARBs reduce podocyte TRPC6 activation and, in turn, reduce expression of TRPC6 by inhibiting angiotensin II-induced calcium influx and calcineurin activation [74]. CNIs directly inhibit NFAT-induced TRPC6 transcription, which reduces TRPC6-induced calcium currents, as discussed above. In addition to these commonly utilized FSGS therapies [121], agonists of protein kinase G (PKG) have been shown to inhibit TRPC6 activity by directly phosphorylating TRPC6 on threonine 69 (Thr69) [122,123,124,125]. PKG is stimulated by activation of either soluble-guanylyl cyclases (sGCs) or particulate-GCs (pGCs) by nitric oxide and natriuretic peptides, respectively [122,123,126]. Moreover, inhibition of TRPC6 by PKG can be potentiated by blocking dephosphorylation of Thr69 by phosphodiesterase (PDE) family members such as PDE5 and PDE9 [125,127,128]. For example, Hall et al. showed that PDE5 inhibition promoted PKG-mediated phosphorylation of Thr69 in cultured podocytes, and negatively regulated TRPC6-mediated calcium conductance [125]. In this study, calcineurin was also shown to dephosphorylate TRPC6 at Thr69, suggesting that the beneficial effects of CNIs may be due, in part, to inhibiting TRPC6 dephosphorylation [125]. In addition, PDE5 inhibition has been shown to negatively regulate TRPC6 expression through cGMP/PKG/PPARγ-mediated transcriptional repression [64]. In a study by Sonneveld et al., treatment with a PDE5 inhibitor reduced proteinuria and glomerular injury in a rat model of FSGS (Adriamycin nephrosis), as well as in a mouse model of diabetic nephropathy [64]. Taken together, these data suggest that repurposing currently approved PDE inhibitors may be useful for the treatment of glomerular diseases by inhibiting both TRPC6 expression and activity. Indeed, PDE5 inhibitors have shown promise in clinical trials for the treatment of patients with established diabetic kidney disease [129].

## 7. Targeting TRPC Family Members to Treat Diabetic Nephropathy

Diabetic kidney disease is the most common cause of ESKD in the United States (U.S.) and Europe, which costs the U.S. ≈20 billion annually [130]. The disease develops in ≈20–40% of patients with diabetes, and progresses to ESKD in ≈20% of patients with overt nephropathy [130]. As a result, there is much interest in developing new therapies.

Similar to FSGS, glomerular podocytes are thought to play a key role in diabetic nephropathy [22,23]. In this regard, podocyte damage and dysfunction are early features of diabetic kidney disease, eventually leading to a decrease in podocyte number and, in turn, disease progression [22,23]. In support of a role for TRPC6 in diabetic nephropathy, glomerular expression of TRPC6 is upregulated in kidneys from rodent models of diabetes [61,64,65,66,67,69], and knockdown of TRPC6 in cultured podocytes both inhibits hyperglycemia-induced apoptosis [90,131] and preserves podocin expression [69]. In addition, several studies suggest that inhibitors of TRPC6 transcription [64,65,72,77] ameliorate albuminuria and the histologic features of diabetic kidney disease in rodent models [64,66,132]. Lastly, hyperglycemia may contribute to kidney injury in diabetes by promoting NOX-dependent ROS generation [133] and, in turn, activation of TRPC6 [41,134]. In support of this possibility, knockout of NOX4 inhibits angiotensin II-stimulated calcium flux in isolated glomeruli from streptozotocin-treated rats, and attenuates diabetic kidney disease in the knockout animals [134].

On the basis of these observations, Spires et al. [112] studied the effect of whole body TRPC6 knockout in Dahl salt-sensitive rats treated with streptozotocin to induce type 1 diabetes. In this study, TRPC6 knockout had no significant effect on hyperglycemia, albuminuria, glomerular injury, interstitial fibrosis, or tubular cast formation. The authors, however, did report that diabetic rats lacking TRPC6 exhibited a reduction in urinary nephrin excretion and a subtle decrease in podocyte injury at the ultrastructural level. On the basis of these data, the authors suggested that TRPC6 inhibition had partial renoprotective effects in diabetic kidney disease.

Our laboratory also examined the effects of whole body TRPC6 knockout in Akita mice, a type 1 model of diabetes [135]. As shown in Figure 5A, we found a reduction in albuminuria early in the disease process, but this difference disappeared as the animals aged [135]. At the end of the study, diabetic TRPC6 knockout exhibited reduced tubular injury, but mesangial expansion was significantly increased (Figure 5B). The adverse effect of TRPC6 knockout on glomerular pathology was associated with insulin resistance that was, at least partially, due to decreased expression of the calcineurin-responsive gene IRS2 (insulin receptor substrate 2), which plays a critical role in insulin signaling. Given that podocytes are insulin-responsive cells [136,137], the adverse effects of TRPC6 knockout on the glomerular compartment may have been exacerbated by podocyte insulin resistance. In support of this possibility, (1) insulin signaling was impaired in TRPC6 knockout podocytes compared to wild type podocytes [135], and (2) previous studies by Welsh et al. [136] found that podocyte specific knockout of the insulin receptor caused heavy proteinuria and histopathologic features of diabetic kidney disease in the normoglycemic environment. Surprisingly, insulin resistance in knockout Akita mice was not associated with changes in blood glucose or hemoglobin A1c levels in our experiments [135]. These observations are, however, consistent with published studies suggesting that insulin sensitizers are beneficial in rodent models of diabetic kidney disease without altering glycemic control [138,139,140]. Taken together with the studies by Spires et al. [112], these data suggest that selective targeting of TRPC6 is unlikely to be an effective treatment strategy in diabetic kidney disease.

Although selective blockade of TRPC6 was only modestly beneficial in diabetic kidney disease, it is possible that inhibition of multiple TRPC family members might be a more effective approach. Indeed, compensatory upregulation of other TRPC family members might compensate for selective inhibition of TRPC6, as discussed above. Liu et al. [141] tested this hypothesis in streptozotocin-treated mice lacking TRPC3, TRPC6, and TRPC7. Despite similar levels of hyperglycemia, knockout mice had decreased glomerular and kidney hypertrophy, reduced albuminuria, and better preserved renal cortical expression of the podocyte marker WT1 (Wilms tumor protein-1). These beneficial effects were associated with a reduction TGFβ (transforming growth factor β) signaling and markers of apoptosis in the knockout mice. Although these findings were impressive, knockout of TRPC3, TRPC6, and TRPC7 markedly reduced body weight in both vehicle-treated and streptozotocin-treated mice, which may have affected the results, as reported in another type 1 diabetic model [142]. Thus, although additional research is needed, currently available studies do not suggest an important therapeutic role for TRPC inhibition in diabetic kidney disease.

## 8. Targeting TRPC6 in Immune-Mediated Glomerular Diseases

In addition to direct effects in the kidney, TRPC6 may influence inflammatory responses [143]. For example, TRPC6 modulates chemotaxis, phagocytosis, cytokine release, and transendothelial migration [143,144,145,146]. To investigate the role of TRPC6 in an immune-mediated injury model, investigators have studied the effects of TRPC6 in nephrotoxic serum nephritis (NTS) [70,71]. In this model, glomerular injury was induced by injection of anti-GBM antibodies [70,71]. A recent study by Dryer and coworkers used the NTS model to examine the effects of TRPC6 knockout in rats [71]. In these studies, the investigators found that deletion of TRPC6 had no significant effect on proteinuria, renal function or tubulointerstitial inflammation and fibrosis compared to wild type animals [71]. In contrast, systemic knockout of TRPC6 significantly reduced glomerulosclerosis and attenuated the decrease in the podocyte marker podocin observed in wild type rats.

In a separate study, Kistler et al. investigated the effects of anti-GBM antiserum in wild type mice and in mice either over-expressing TRPC6 specifically in podocytes or in TRPC6 knockout mice [70]. Unexpectedly, these investigators found that podocyte-specific overexpression of TRPC6 attenuated albuminuria and FP effacement in NTS. In contrast, proteinuria was enhanced in the early stages of the disease process in TRPC6 knockout mice, and this increase in albuminuria was significantly reduced by complement depletion prior to induction of NTS. The authors attributed the beneficial effects of TRPC6 overexpression to enhanced activation of CaMKII on the basis of the following observations: (1) increased intracellular calcium levels protected podocytes from complement-mediated cellular injury [147]; (2) pharmacologic blockage of CaMKII exacerbated complement-mediated cellular injury in cultured podocytes [70]; and (3) CaMKII activation was decreased in TRPC6 knockout mice, but was increased in mice overexpressing TRPC6 specifically in podocytes [70]. In this study, the effects of TRPC6 knockout on glomerulosclerosis were not specifically evaluated, and the downstream effector pathways of CaMKII were not further explored. However, the differing effects of TRPC6 knockout in animal models of NTS, FSGS, and diabetic kidney disease suggest that targeting TRPC6 to treat glomerular disease processes is likely to be disease-specific.

## 9. Targeting TRPC5 in Proteinuric Kidney Diseases

As discussed above, Rho GTPase family members such as RhoA, Rac1, and Cdc42 play key roles in glomerular disease processes [107]. Excessive activation or loss of Rho GTPase family members causes proteinuria and effacement of podocyte foot processes in animal models [103,105,107,148,149,150,151,152]. Moreover, Rho GTPases promote podocyte loss by inducing podocyte apoptosis [152] and podocyte detachment [105]. Thus, targeting small GTPases for the treatment of glomerular disease processes is an active area of investigation.

A role for TRPC channels in regulating podocyte Rho GTPase activity was suggested by Greka and colleagues [100]. These investigators found that TRPC6 and TRPC5 were coupled to activation of RhoA and Rac1, respectively, and regulated actin cytoskeletal dynamics [100]. The authors suggested that TRPC6 promotes stress fiber formation and a contractile phenotype, which is essential for maintaining normal filtration barrier integrity [100,101]. In contrast, TRPC5 inhibited stress fiber formation and reduced podocyte contractility, which the authors suggested may play a role in adapting to hydrostatic stresses in the glomerulus [100]. In this scenario, the coupling between TRPC5 and Rac1 is of interest because Rac1 induces rapid translocation and insertion of TRPC5 into the cell membrane and increases calcium influx [153]. This creates a positive feedback loop with TRPC5, inducing Rac1 activation and, in turn, Rac1 promoting enhanced TRPC5 activity [100,101]. Greka and colleagues hypothesized that unopposed and excessive Rac1 signaling might play a role in glomerular diseases by promoting maladaptive cytoskeletal remodeling and induce proteinuria [101]. In support of this hypothesis, TG expression of a constitutively active Rac1 construct in podocytes induces proteinuria, FP effacement, glomerulosclerosis, and excretion of podocytes expressing the transgene in urine [105,151].

To determine if TRPC5 played a pathogenetic role in proteinuric kidney diseases, Greka and coworkers [108] examined the effects of TRPC5 inhibition in animal models of proteinuria and FP effacement. In these experiments, whole body knockout of TRPC5 inhibited both albuminuria and effacement of podocyte FPs in the lipopolysaccharide (LPS) model of proteinuria. Similarly, knockout of TRPC5 inhibited FP effacement using the ex vivo, protamine sulfate (PS) model of glomerular filtration barrier disruption. The investigators then examined pharmacologic inhibition of TRPC5 using the combined TRPC4 and TRPC5 inhibitor ML204 [154]. This compound is selective for TRPC4 and TRPC5 [154], and data from this research group indicated that TRPC4 was not a major contributor to angiotensin II-induced calcium entry in podocytes [100]. On the basis of these data, the investigators suggested that examining TRPC channel activity using ML204 selectively examines the contribution of TRPC5 to podocyte biology. Indeed, the investigators found that ML204 had effects similar to TRPC5 knockout using both the LPS and PS models. On the basis of these observations, the researchers hypothesized that TRPC5 inhibition might be a useful strategy for treatment of proteinuric kidney disease.

To further investigate this hypothesis, Greka and co-workers examined the effects of ML204 in a more aggressive proteinuria model induced by TG overexpression of the human type 1 angiotensin II receptor (AT1R) specifically in podocytes in rats (AT1R TG rats) [155]. In these studies, ML204 inhibited albuminuria and podocyte loss, and improved glomerular histology in the AT1R TG model, even when treatment began in rats after the onset of disease [109]. To more selectively examine the role of TRPC5 in the disease process, the investigators developed a specific TRPC5 inhibitor (AC1903) based on the structures of published TRPC5 inhibitors including the TRPC4/5 inhibitor ML204 [154,156]. In these studies, AC1903 suppressed proteinuria, reduced podocyte loss, and improved glomerular histology in the in AT1R TG rats [109]. Similar beneficial effects of AC1903 were also observed in Dahl salt-sensitive rats, which are a model of hypertension-induced glomerulosclerosis. Thus, the data support a role for TRPC5 in the pathogenesis of proteinuric kidney diseases, provided that the doses of the pharmacologic agents maintained selectivity for TRPC5 in vivo.

Despite these promising findings, the role of TRPC5 in glomerular disease processes is controversial. Reiser and colleagues [157] studied the effects of overexpressing either wild type TRPC5 or a dominant-negative TRPC5 using a promoter that expressed the TG TRPC5 constructs ubiquitously [158]. The investigators found that both TG and non-TG mice exhibited similar levels of proteinuria as the mice aged, as well as after treatment with LPS. Furthermore, injection of the TRPC5 activator englerin-A did not cause proteinuria in non-TG mice or in TG animals, and treatment with ML204 did not inhibit LPS-induced albuminuria in any of the groups. The authors suggested that TRPC5 does not cause or exacerbate glomerular injury. However, interpretation of this study is complicated. First, primary podocyte cultures from animals overexpressing wild type TRPC5 demonstrated higher calcium responses to carbachol compared to podocytes overexpressing the dominant negative construct; however, no comparison of calcium responses was made with primary podocytes from non-TG mice. As a result, it is not known if the TG mice exhibited significantly altered calcium responses compared to non-TG animals. In addition, although the dose englerin-A used in these studies was biologically active in other model systems [159,160,161], the treatment regimen may not have been of sufficient strength or duration to cause adverse renal effects. Lastly, the dosage of ML204 used in the studies by Reiser and colleagues [157] was ≈10-fold lower than the dosage used in the studies by Greka and coworkers [108,109]. Thus, the dosage may have been too low to be effective in the studies by Reiser and colleagues [157] or, alternatively, the dosage used by Greka and coworkers [108,109] may have had beneficial effects due to blockade of other TRPC family members.

On the basis of these publications, further studies will be necessary to elucidate the role of TRPC5 in kidney diseases. In this regard, TRPC6 is the major TRPC family member linked to GPCR-coupled calcium entry in podocytes under basal conditions [41,43,162]. In disease states, however, other TRPC family members may play a role in disease pathogenesis. For example, active TRPC5 channels are difficult to detect prior to disease onset in the AT1R TG rat described above [109]. In contrast, with the onset of disease, functional TRPC5 channels are readily detected, and their activity may increase further as the disease progresses [109]. The mechanism(s) of enhanced TRPC5 activity in this model was not further investigated in this published study, but changes in the level of TRPC5 expression in the kidney have not been observed in animal models of glomerular disease [61,71,73,111,112]. However, the level of channel activity is also modulated by the number of functional channels that traffic to the cell surface [163]. This observation is directly relevant to TRPC5 channels, because cell surface expression of TRPC5 may be regulated by some of the same stimuli that activate TRPC6, including mechanical stimuli and oxidative stress [31,32,38,39]. In this scenario, both TRPC5 and TRPC6 might traffic to the cell surface in a coordinated fashion in disease states and, in turn, contribute calcium entry and podocyte injury in proteinuric kidney diseases.

## 10. Targeting TRPC Family Members in Other Acquired Kidney Diseases

In addition to podocytes, TRPC6 is expressed in multiple cell types in the kidney [35]. Recent studies suggest an important role for the TRPC family members TRPC3 and TRPC6 in fibroblast proliferation, as well as in fibroblast to myofibroblast transformation [75,164]. These cell types play a critical role in wound healing and are the key cellular mediators of fibrosis in renal and extrarenal tissues [75,76,164]. Saliba et al. found that selective pharmacologic blockade of TRPC3 using Pyr3 [165] inhibited fibroblast proliferation, myofibroblast transformation, and production of extracellular matrix proteins in primary renal fibroblasts [75]. In vivo, treatment with Pyr3 inhibited fibroblast activation, attenuated both interstitial fibrosis and inflammatory cell infiltration, and decreased total renal tissue collagen content in obstructed kidneys of both Wistar rats and wild type mice undergoing unilateral ureteral obstruction (UUO) [75]. Pharmacologic blockade of TRPC3 also inhibited upregulation of TRPC3 in obstructed kidneys, and the beneficial histologic effects of TRPC3 inhibition were recapitulated in TRPC3 knockout mice.

In a separate study, Wu et al. found that TRPC6 knockout provided partial protection from renal fibrosis in mice subjected to UUO, and a comparable beneficial effect was observed in obstructed kidneys following treatment with the non-selective TRPC inhibitor BTP2 [76]. Moreover, a similar beneficial effect was observed by another group in a mouse UUO model using the selective TRPC6 inhibitor BI-749327 [166]. In the Wu et al. study, the authors also examined the effects of soluble klotho in the UUO model on the basis of the following observations: (1) the investigators had previously demonstrated that soluble klotho inhibits TRPC6 exocytosis and TRPC6 currents in the heart [167], and (2) treatment with soluble klotho inhibits renal fibrosis in obstructed kidneys [168]. However, soluble klotho provided no additional protection in TRPC6 knockout mice, suggesting that the beneficial effects of klotho in UUO were mediated by inhibition of TRPC6. Lastly, both TRPC3 and TRPC6 mRNA were potently upregulated in interstitial fibroblasts of obstructed kidneys [76], but combined TRPC3 and TRPC6 knockout in UUO provided no additional protection from renal fibrosis compared to TRPC6 knockout mice. Given that either TRPC3 or TRPC6 knockout attenuated fibrosis in UUO, it is of interest that combined TRPC3 and TRPC6 knockout was not additive in the Wu et al. study [76]. We can only speculate on the mechanisms of this effect; however, TRPC1, TRPC3, and TRPC6 co-immunoprecipitate in freshly isolated renal fibroblasts, suggesting that these TRPC family members form a macromolecular complex in these cells [75]. Moreover, TRPC6 has been shown to form heterodimers with TRPC3 in other cell types [35,169]. These results suggest that TRPC6 and TRPC3 are components of the same signaling pathways, perhaps by forming heterodimers or macromolecular complexes with other TRPC family members.

## 11. Conclusions

In summary, the discovery that gain-of-function mutations in TRPC6 caused familial forms of FSGS generated significant interest in targeting TRPC6, and perhaps other TRPC family members, to treat glomerular disease processes. Since that time, preclinical studies have examined the role of TRPC channels in diverse animal models of kidney disease. Additional studies have suggested that TRPC6 is a component of a novel signaling complex at the SD that likely plays a key role in podocyte biology. Interpretation of available studies is, however, limited by differing results between disease models, as well as generalizability of preclinical results to human disease processes. Although we acknowledge the limitations of preclinical models, these model systems have significantly advanced our understanding of both normal physiology and pathophysiologic processes. Moreover, preclinical models will continue to play an important role in developing molecular-targeted therapies for human disease. In this regard, a major limitation of current model systems is the lack of inducible, cell-specific TRPC knockout models. Conditional knockout models would enhance our ability to discern the biologic role of TRPC family members in organs and tissues composed of multiple, diverse cell types such as the kidney. The study of glomerular biology is also limited by current cell culture systems, which cannot recapitulate the multicellular environment and hydrostatic pressures of the glomerulus. Recent progress in cell culture systems may circumvent some of these limitations. For example, newly developed microfluidic devices are able to support the growth of multicellular culture systems, which closely simulate the natural tissue–tissue interface of the glomerulus [170]. It is likely that the development of both new animal models and innovative cell culture technologies will facilitate the study of glomerular biology. Given the need for new therapies to treat kidney diseases, the development of these new technologies provides an opportunity to better understand normal physiology and pathophysiologic processes, as well as to test new therapeutic strategies.

## Figures and Tables

**Figure 1 cells-09-00044-f001:**
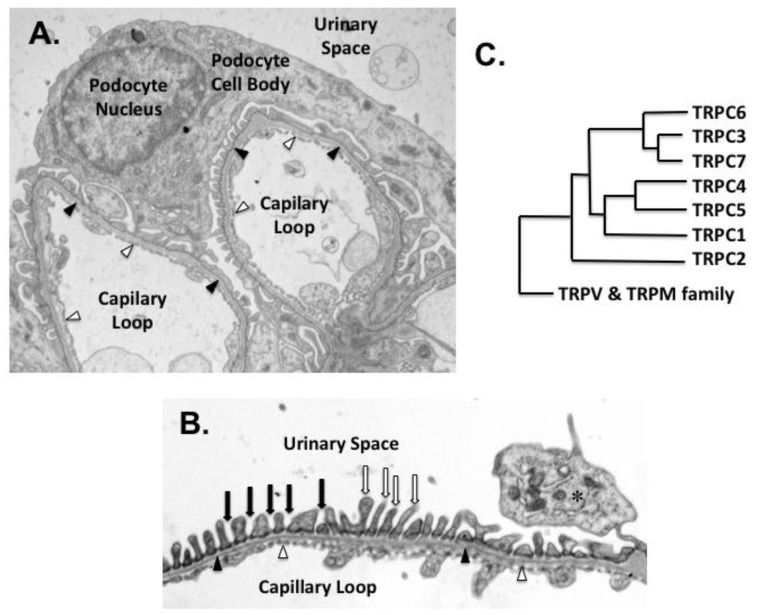
Glomerular ultrastructure and TRPC family members. (**A**) An electron micrograph of glomerular ultrastructure is presented. The podocyte cell body is located in the urinary space on the outer surface of the glomerular basement membrane (GBM). Primary processes extend from the cell body and branch into interdigitating foot processes (FPs) that cover the external surface to the GBM (black arrow heads) to form filtration slits and the specialized intercellular junction termed the slit diaphragm (SD). The glomerular filter is composed of the podocyte FPs with the intervening SDs, the GBM (black arrow heads), and a fenestrated endothelium (white arrowheads) that lines the capillary loops. (**B**) This high power view shows the three layers of the glomerular filter including the fenestrated endothelium (white arrowheads), GBM (black arrowheads), and the FPs (white arrows) with the interposed filtration slits (black arrows). An asterisk indicates a primary process (also termed a major process). (**C**) The phylogenetic tree of TRP family members is shown in panel C. The vertebrate TRPC family has seven members, which are divided into four subgroups: TRPC1, TRPC2, TRPC4/5, and TRPC3/6/7. The figure shows the relationship between TRPC family members and other members of the TRP family. In humans, TRPC2 is a pseudogene (see text).

**Figure 2 cells-09-00044-f002:**
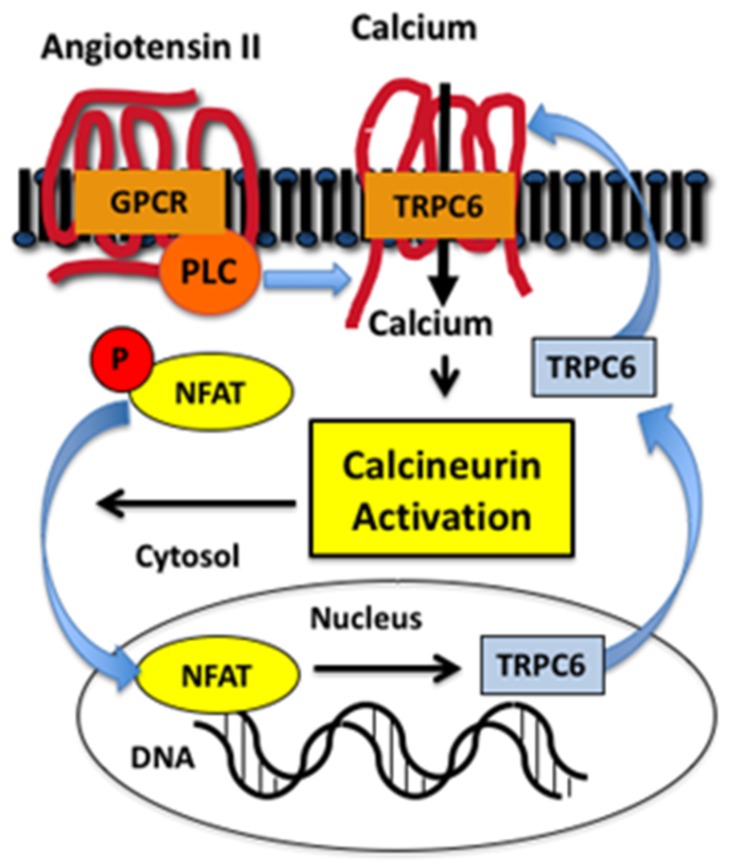
Positive feedback loop induced by TRPC6 activation. Stimulation of phospholipase C (PLC)-coupled receptors such as the G protein coupled receptor (GPCR) for angiotensin II activates TRPC6, which induces calcium entry into the cell, calcineurin activation, dephosphorylation of nuclear factor of activated T cells (NFAT), and translocation of NFAT to the nucleus. NFAT stimulates transcription of multiple genes including TRPC6, which further increases TRPC6 expression and enhances the calcium influx, creating a positive feedback loop.

**Figure 3 cells-09-00044-f003:**
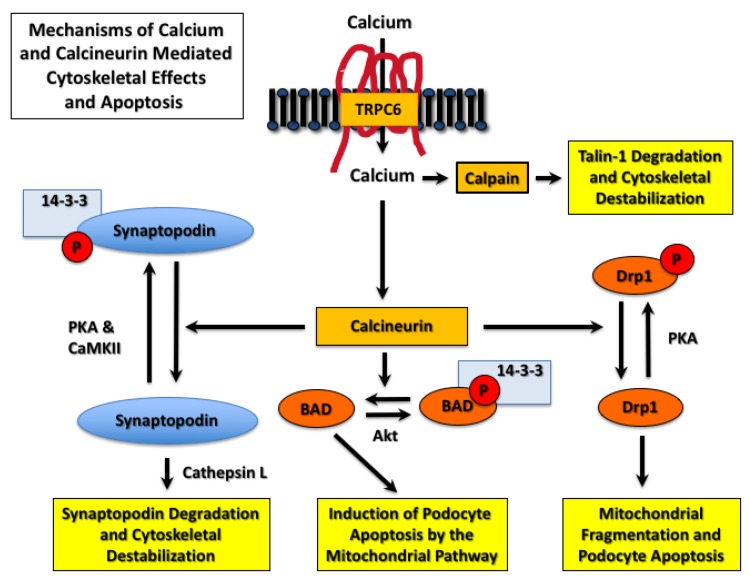
Mechanisms of calcium- and calcineurin-mediated podocyte injury. The podocyte protein synaptopodin is an actin associated protein that plays a role in stabilizing the podocyte cytoskeleton. Synaptopodin is phosphorylated by protein kinase A (PKA) and calcium/calmodulin-dependent protein kinase II (CaMKII), which promotes 14-3-3 binding and protects synaptopodin from degradation by cathepsin L. Calcineurin dephosphorylates synaptopodin, which permits its degradation by cathepsin L and, in turn, destabilizes the actin cytoskeleton and promotes proteinuria. Calcineurin also causes podocyte apoptosis. The mechanisms of calcineurin-induced apoptosis are incompletely understood, but both BAD (Bcl-2-associated death promoter) and Drp1 (dynamin-related protein 1) play a role in preclinical disease models. Similar to synaptopodin, BAD is phosphorylated at a 14-3-3 docking site by Akt, which inhibits its activity. Calcineurin dephosphorylates BAD and induces apoptosis by the mitochondrial pathway. Drp1 is phosphorylated and inhibited by PKA. Dephosphorylation of Drp1 by calcineurin promotes apoptosis by inducing mitochondrial fragmentation. Lastly, recent studies suggest that the calcium-activated cysteine protease calpain plays a key role in promoting podocyte injury. Calpain degrades numerous cytoskeletal proteins including the large cytoskeletal protein talin-1. Calpain-induced talin-1 degradation destabilizes the podocyte cytoskeleton and impairs glomerular filtration barrier integrity.

**Figure 4 cells-09-00044-f004:**
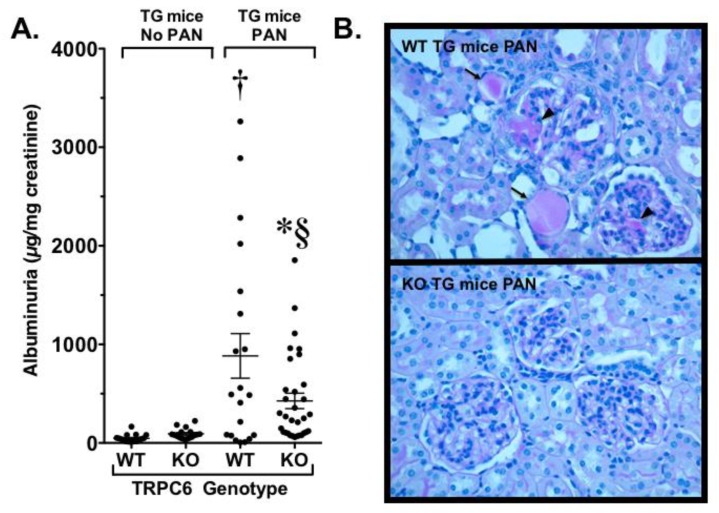
Effect of TRPC6 knockout (KO) in an animal model of focal segmental glomerulosclerosis (FSGS). Our lab developed a transgenic (TG) mouse model that promoted continuous activation of endogenous TRPC6 specifically in podocytes. This model had no kidney phenotype at baseline. In contrast, the podocyte toxin puromycin aminonucleoside (PAN) induced heavy proteinuria in TG mice but only mild proteinuria in non-TG animals (see text). Using this TG model, we found that whole-body knockout of TRPC6 attenuated albuminuria compared to wild type (WT) TG mice expressing TRPC6 (**A**), and reduced both segmental glomerulosclerosis (arrowheads) and tubular injury (arrows) in the knockout animals (**B**). * *p* < 0.05 vs. WT TG mice treated with PAN, † *p* < 0.005 vs. WT TG mice at baseline (no PAN), § *p* < 0.005 vs. KO TG mice at baseline (no PAN).

**Figure 5 cells-09-00044-f005:**
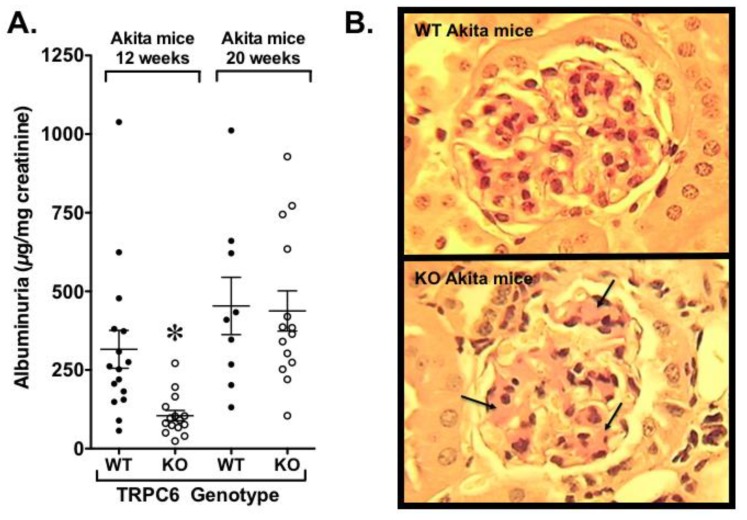
Effect of TRPC6 knockout in an animal model of diabetic kidney disease. Using a model of type 1 diabetes mellitus (Akita mice), we found that whole-body knockout of TRPC6 attenuated albuminuria early in the disease process (12 weeks of age), but this difference disappeared by 20 weeks of age (**A**). Histologic examination of kidneys was performed at the 20 week time point. Knockout of TRPC6 enhanced mesangial expansion (arrows) in knockout Akita mice compared to wild type (WT) Akita mice expressing TRPC6 (**B**). * *p* < 0.05 vs. WT Akita mice at 12 weeks of age.

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
