# Peer review of "TRPC Channels in Proteinuric Kidney Diseases"

_cells, 2019, doi:10.3390/cells9010044_

Round 1
Reviewer 1 Report
This is a timely review written by experts in the field. The article summarizes the data about members of the TRPC family in the kidney and their role in proteinuric kidney diseases. The work is noteworthy and authoritative. Importantly, it is well balanced and discusses various opinions about contributions of TRPC channels in the kidney in unbiased way. The manuscript is well written and I recommend its publication. I have a few minor comments that I hope will improve this work. However, all these suggestions are up to the authors consideration.
1) Title – I believe “cation” could be removed.
2) Add somewhere that TRPC channels sometimes also called “Transient Receptor Potential Canonical (TRPC) channels”.
3) Keywords: “transient potential receptor cation channel” – should be “transient receptor potential cation channel”
4) line 130 – remove extra “in”
5) line 171 – “present presentation” – rephrase
6) line 300 – Reference #75 might not be needed here since as described in an abstract “The magnitude of protection against obstruction-induced fibrosis in Trpc3 and Trpc6 double knockout mice was not different from that in Trpc6 knockout mice.”
7) Please consider to describe contribution of Nox4 in the regulation of Trpc6 in podocytes (PMID: 29793963).
8) Consider to describe recent study demonstrating increased expression of TRPC6 and decreases in the expression of podocin and nephrin in STZ rats and cultured podocytes (PMID: 31566428).
9) Lines 569-570: Spires et al also attempted to estimate the level of TRPC5 in STZ-injected Dahl SS rats and in contrast to Trpc6, Trpc5 levels did not change neither in SSTrpc6-/- rats nor WT rats following STZ treatment.
10) Please discuss, if any information available, compensatory upregulation of TRPC3.
Author Response
Reviewer#1
1) Title – I believe “cation” could be removed.
We agree with this suggestion and "cation" has been removed from the title.
2) Add somewhere that TRPC channels sometimes also called “Transient Receptor Potential Canonical (TRPC) channels”.
We acknowledge that TRPC channels are also termed “Transient Receptor Potential Canonical (TRPC) channels” (see line 95-96 in the revised manuscript).
3) Keywords: “transient potential receptor cation channel” – should be “transient receptor potential cation channel”
The key word “transient potential receptor cation channel” has been corrected.
4) line 130 – remove extra “in”
5) line 171 – “present presentation” – rephrase
Thank you for these corrections.
6) line 300 – Reference #75 might not be needed here since as described in an abstract “The magnitude of protection against obstruction-induced fibrosis in Trpc3 and Trpc6 double knockout mice was not different from that in Trpc6 knockout mice.”
We have deleted reference #75 in line 300 of the original manuscript (line 300 and reference #76 in the revised manuscript.
7) Please consider to describe contribution of Nox4 in the regulation of Trpc6 in podocytes (PMID: 29793963).
We have described the contribution of Nox4 to regulation of TRPC6 in the revised manuscript [lines 421-425 in the revised manuscript].
8) Consider to describe recent study demonstrating increased expression of TRPC6 and decreases in the expression of podocin and nephrin in STZ rats and cultured podocytes (PMID: 31566428).
We added the reference [#69] to the revised manuscript and described the effect of TRPC6 knockout on podocin expression [see lines 418-419 of the revised manuscript]
9) Lines 569-570: Spires et al also attempted to estimate the level of TRPC5 in STZ-injected Dahl SS rats and in contrast to Trpc6, Trpc5 levels did not change neither in SSTrpc6-/- rats nor WT rats following STZ treatment.
We have added this reference [#112] to lines 576-577 in the revised manuscript.
10) Please discuss, if any information available, compensatory upregulation of TRPC3.
Compensatory upregulation of TRPC3 is discussed in lines 307-313 and 454-455 in the revised manuscript.
Reviewer 2 Report
Overall, the review is comprehensive and up-to-date on the subject of the potential role of TRPC5 and TRPC6 in human kidney diseases, animal models of kidney disease, and the potential for pharmacologic inhibitors of these channels in the treatment of chronic kidney disease. I have only a few minor suggestions.
Figure 1: A higher magnification image of B should be shown to better allow visualization of the endothelium, GBM and foot processes with SDs. Page 3, line 110: whether TRPC channels, including TRPC6, are in fact responsive to mechanical stretch remains controversial. A recent paper (PMID: 31722978) argues that these channels as a group, including C6, are not mechanosensitive. Figure 3: as shown, the figure implies that phosphorylated Bad induces apoptosis, which would suggest that calcineurin activity is anti-apoptotic. However, in the figure legend, the opposite is implied (that dephosphorylated Bad generated by calcineurin activity is pro-apoptotic). Please clarify this point. This reviewer feels that the current literature provides conflicting conclusions regarding the potential benefit or utility of blocking TRPC6 or TRPC5 in kidney disease, depending on the animal and disease model examined. Coupled with the uncertainty as to the human corollaries of a number of these disease models, the role of TRPC6 in human renal disease beyond the genetic forms remains very uncertain. While this is made apparent in the body of the review, it should be stressed again in the conclusions section of the manuscript. Proof-reading for grammatical errors is needed.Author Response
Reviewer #2
Figure 1: A higher magnification image of B should be shown to better allow visualization of the endothelium, GBM and foot processes with SDs.
We have increased the size of Figure 1B in the revised manuscript to enhance visualization of the endothelium, GBM and foot processes with SDs. Unfortunately, enlargement is limited by resolution of currently available electron microscopic images.
Page 3, line 110: whether TRPC channels, including TRPC6, are in fact responsive to mechanical stretch remains controversial. A recent paper (PMID: 31722978) argues that these channels as a group, including C6, are not mechanosensitive.
Thanks for the suggestion We have added the reference and commented on the controversial role of mechanical stretch in activation TRPC family members in the revised manuscript (lines 115-116).
Figure 3: as shown, the figure implies that phosphorylated Bad induces apoptosis, which would suggest that calcineurin activity is anti-apoptotic. However, in the figure legend, the opposite is implied (that dephosphorylated Bad generated by calcineurin activity is pro-apoptotic). Please clarify this point.
The mistake in Figure 3 has been corrected to reflect dephosphorylation of Bad by calcineurin and generation of its pro-apoptotic form.
This reviewer feels that the current literature provides conflicting conclusions regarding the potential benefit or utility of blocking TRPC6 or TRPC5 in kidney disease, depending on the animal and disease model examined. Coupled with the uncertainty as to the human corollaries of a number of these disease models, the role of TRPC6 in human renal disease beyond the genetic forms remains very uncertain. While this is made apparent in the body of the review, it should be stressed again in the conclusions section of the manuscript.
We agree with the reviewer that the current literature provides conflicting conclusions regarding the potential benefit of targeting TRPC family members in human disease, in part, because of differing results between disease models and generalizability of preclinical results to human disease processes. We have discussed these limitations in the Conclusions Section of the revised manuscript (lines 622-623).
Proof-reading for grammatical errors is needed.
We apologize for the grammatical errors, and we have attempted to correct these mistakes in the revised manuscript.